# Responses to Sensory Events in Daily Life in Children with Cerebral Palsy from a Parent Reported Perspective and in a Swedish Context

**DOI:** 10.3390/children10071139

**Published:** 2023-06-30

**Authors:** Annika Ericson, Åsa Bartonek, Kristina Tedroff, Cecilia Lidbeck

**Affiliations:** 1Department of Women’s and Children´s Health, Karolinska Institutet, S-171 76 Stockholm, Sweden; asa.bartonek@ki.se (Å.B.); kristina.tedroff@ki.se (K.T.); cecilia.lidbeck@ki.se (C.L.); 2Neuropediatric Unit, Astrid Lindgren Children’s Hospital, Karolinska University Hospital, S-171 76 Stockholm, Sweden

**Keywords:** behaviour, cerebral palsy, Child Sensory Profile-2, motor functioning, posture, sensory processing, sound sensitivity

## Abstract

The motor disorders of cerebral palsy (CP) are often accompanied by sensory disturbances, but knowledge of their relationship to motor functioning is sparse. This study explored responses to sensory events in relation to spastic subtype and motor functioning in children with CP. Parents of 60 children with CP (unilateral: 18, bilateral: 42) with GMFCS levels I:29, II:13, III:15 and IV:3 of mean age 12.3 years (3.7 SD) participated. The parents (*n* = 55) rated their children´s responses with the norm-referenced questionnaire Child Sensory Profile-2© (CSP-2©), Swedish version, incorporating nine sections and four sensory processing patterns/quadrants, and replied (*n* = 57) to two additional questions. On the CSP-2©, thirty (55%) of the children were reported to have responses “much more than others“ (>2 SD) in one or more of the sections and/or quadrants and 22 (40%) in the section of Body Position, overrepresented by the children with bilateral CP. The additional questions revealed that a greater proportion of children at GMFCS levels III-IV compared to level I frequently were requested to sit/stand up straight (14/17 versus 6/26, *p* < 0.001) and were sound sensitive at a younger age (14/17 versus 10/26, *p* = 0.005). The findings of this study highlight the sensory aspects of motor functioning in children with spastic CP.

## 1. Introduction

Sensory processing can be described as the central nervous system’s ability to organize and interpret multiple sensory modality inputs from the body and the environment to produce an adaptive response [1,2]. It is a complex process that is important for children’s participation and functioning in daily life [3]. Perception, defined as specific mental functions of recognizing and interpreting sensory information [4], and action are mutually dependent on each other [5]. For children with cerebral palsy (CP) with activity limitations caused by disturbances in the developing foetal or infant brain [6], this raises a challenge. Cerebral palsy is described as a disorder of movement and posture with accompanying disturbances such as of sensation, perception, cognition and behaviour [6]. In the definition of CP from 2007, disturbances of sensation and perception are explained both as a function of the primary disorder and as a consequence of the activity limitations [6]. The presence of impairments in modulating sensory input in children with CP is also in line with the fact that many children have neuropsychiatric impairments such as autism spectrum disorder (autism) [7,8], which can alter sensory behaviours [9]. The motor impairments in CP are most frequently evaluated in accordance with internationally well-established instruments. Motor functioning can be classified using the five levels of the Gross Motor Function Classification System (GMFCS) Expanded and Revised version, ranging from level I, walking without limitations, to level V, being transported in a wheelchair [10].

Cerebral palsy is classified into different subtypes based on the predominant neurological symptoms [11]. With respect to distribution, the spastic subtype can be divided into unilateral spastic CP (US-CP), involving limbs on one side of the body, and bilateral spastic CP (BS-CP), involving limbs on both sides of the body [11]. In children with US-CP, manual ability is often more limited than gross motor function [12], with functioning commonly at GMFCS levels I-II [7,13]. Bilateral spastic cerebral palsy is the most common subtype of CP and includes children at all GMFCS levels [14,15]. Sensory disturbances have been described in children with cerebral palsy; these include difficulties with tactile sensibility in the hand affecting object discrimination [16], proprioception affecting postural stability and gait speed [17] and visual–perceptual impairments [18] affecting visuo-motor integration such as eye–hand coordination and praxis-constructional abilities [19].

Some studies have investigated the importance of sensory information for postural control in CP, often with the use of three-dimensional motion analysis and sometimes in combination with superficial electromyography [20,21,22,23]. Results show that children with BS-CP often stand with flexed knees and have difficulty maintaining their body position in relation to gravity [24]. They are more dependent on vision to orient their posture during standing [20] and walking [25] compared to typically developing children. Motor and perceptual disorders may co-exist in children with CP and can lead to insecurity and anxiety during moving when stability limits are provoked [26].

Perceptual disturbances entailing difficulty analysing information coming from surrounding space, such as visual and auditory stimuli and information from movements and body position in relation to gravity have been observed in children with BS-CP [26,27]. In some children with BS-CP, sudden stimuli, such as unexpected sounds, may elicit physical reactions such as the startle reaction, frequent eye blinking and/or grimacing. These reactions have been described as clinical signs of a perceptual disorder [26]. Some children with BS-CP with a perceptual disorder affecting movement have difficulty automatically correcting their posture in relation to gravity, although they can assume an erect body posture when requested. These difficulties have been related to suppression of the analysis of sensory information [28]. Knowledge of how sensory disturbances and difficulties with sensory processing relate to postural control and motor functioning in daily life for children with CP is sparse [29,30]. Neuroimaging and neurophysiology techniques have enhanced the knowledge of how the central nervous system organizes and interprets sensory inputs from the body and the environment [31,32,33]. Moreover, self- and proxy-reports, such as questionnaires, have gained new insight into how individuals perceive and handle themselves in relation to the task and the environment and manage to produce adaptive responses and actions.

Sensory processing, measured as observed behavioural responses to sensory events in daily life, has been investigated in children with CP in a few studies [34,35,36] with a caregiver questionnaire, the Sensory Profile [37,38]. Children with spastic CP were reported to differ in their responses compared to typically developing children [34,35,36]. To our knowledge, there are no previously published studies measuring responses to sensory events with the Sensory Profile in children with CP in a Swedish context.

The aim of this study was to explore parents’ reports of how their child with CP responds to sensory events in daily life. We hypothesized that the children with CP would differ in their responses compared to children with typical development. Moreover, we hypothesized that the children with BS-CP would differ in their responses compared to the children with US-CP and with respect to level of motor functioning.

## 2. Materials and Methods

### 2.1. Participants

In this cross-sectional study, the parents of 60 children with spastic CP participated. Inclusion criteria were sufficient language skills in Swedish to answer the study questions and their children being 6–18 years of age and functioning at GMFCS levels I-IV. The parents were recruited through the Karolinska University Hospital in Stockholm, Sweden in conjunction with scheduled outpatient visits of their children to the departments of neuro-paediatrics between April 2019 and August 2020. Prior to participation, all children gave verbal assent, and the parents gave written consent after being given oral and written information regarding the study protocol. The study was conducted in accordance with the Declaration of Helsinki and approved by the Regional Ethics committee in Stockholm and the Swedish Ethical Review Authority (2018/1841-31: 24 October 2018, 2019-01372: 7 March 2019).

### 2.2. The Child Sensory Profile-2© (CSP-2©)

The caregiver questionnaire Child Sensory Profile-2© (CSP-2©) [38], Swedish version (Sensory Profile 2 Copyright © 2014 NCS Pearson, Inc. Swedish translation copyright © 2016 NCS Pearson, Inc. Adapted and reproduced by Pearson Sweden AB under license from NCS Pearson, Inc. All rights reserved. Sensory Profile 2 results are published with permission from Pearson Sweden AB. Pearson has not been involved in the study and is neither responsible for the quality of the results nor for the overall outcome of the conducted research) [39] was used to measure the children´s rate of responses to sensory events in daily life and to document their sensory processing patterns. The Sensory Profile is an established caregiver questionnaire and can be used as a functional measure of sensory responsivity or to gather data linking the impact of sensory processing on participation in daily activities [38]. The instrument was designed for all children, although in research, its use has commonly targeted children with autism and attention-deficit/hyperactivity disorder (ADHD) [40]. The Sensory Profile© has several versions, and the Child Sensory Profile-2© (CSP-2©) [38] is one of the more recently developed versions. The CSP-2© has proven strengths in patient-related outcome measure design with merit in internal consistency, content- and construct validity for children 3:0–14:11 years of age [38,40]. The CSP-2© questionnaire consists of 86 items/statements structured in six sensory and three behavioural sections that generate separate scores. Scores from items in the sections are transferred to calculate quadrants, which intend to describe the child’s sensory processing patterns (Figure 1).

The conceptual framework of the quadrants, named Dunn´s Sensory Processing Framework [38], suggests an interaction between the neurological threshold to sensory input, described as the amount of stimuli required for a neural response, and the child´s self-regulating behaviour, described as the child’s ability to handle their own needs [38]. In the CSP-2© questionnaire, the parents report how frequently their child is responding in the described manner on a Likert-like scale ranging 5–0: always (90% or more of the time), frequently (75% of the time), half the time (50% of the time), occasionally (25% of the time), almost never (10% or less of the time) or does not apply. High scores on the CSP-2© are associated with frequent behaviours, whereas low scores are associated with less frequent behaviours [38]. The CSP-2© is a norm-referenced questionnaire and the Swedish norm data is based on 729 reports from Scandinavian parents [39]. Individual scores for both sections and quadrants are described in relation to norm-referenced standard deviations (SD). The “more/less than others” (±1–2 SD) and “much more/much less than others” (±2 SD) are used to represent scores outside of the majority/norm. Scores deviating ±2 SD from the norm-referenced values indicate a definite difference in responses, which may affect the child´s participation in activities of daily life [38,41].

### 2.3. Questions Regarding Maintaining Body Posture and Sound Sensitivity

Two questions regarding the ability to maintain an erect body posture and sensitivity to sounds were formulated, based on the researchers’ professional experiences from the group of patients and previous research [20,28]. The questions were: (a) ‘Is your child frequently requested to sit/stand up straight?’ and (b) ‘Was your child sensitive to sounds at a younger age?’ The questions were answered in writing (yes/no) by the parents.

### 2.4. Procedures

During a physical visit, parents replied to the CSP-2© [38], Swedish version [39], and the two additional questions. An experienced paediatric physical therapist was present for guidance if needed.

### 2.5. Data and Statistical Analysis

Descriptive data on age, sex, subtype of spastic CP, GMFCS level, scores on CSP-2© and answers to the questions regarding maintaining body posture and sound sensitivity are presented as mean (SD), median [min, max], numbers and/or percent. The children functioning at GMFCS level III and those at level IV were grouped together (level III-IV). Correlations were calculated between age and scores on the CSP-2© with Spearman’s correlation coefficient (r_s_). Correlations were considered significant for *p* ≤ 0.05 and r_s_ ≤ −0.30 or ≥0.30. The following interpretation was used for the size of the agreement r_s_: 0.00–0.30, little if any correlation; 0.30–0.50, low correlation; 0.50–0.70, moderate correlation; 0.70–0.90, high correlation; and 0.90–1.00, very high correlation [42]. Non-parametric statistics, the Mann Whitney U test and the Kruskal Wallis test, were used to analyse possible differences in median [min, max] scores on the CSP-2© with respect to CP subtype and GMFCS levels. A two-sided Pearson Chi-Square test was used to compare proportions of children with scores > 2 SD in the sections of the CSP-2© and proportions of parents’ answers on the questions regarding maintaining posture and sound sensitivity with respect to subtype and GMFCS levels using SPSS (version 27, SPSS Inc., Chicago, IL, USA). Parametric statistics, namely a two-sided Z-test, was used to analyse possible differences on CSP-2© mean scores between the entire group of children with CP, the children with US-CP and the children with BS-CP compared to norm-referenced values using R (version 4.1.2 and 4.2.0). The significance level was set at *p* ≤ 0.05.

## 3. Results

### 3.1. Participants

Participants were parents of 60 children with spastic CP representing 38 (63%) boys and 22 (37%) girls (*p* < 0.001), mean (SD) age 12.3 (3.7) years, functioning at GMFCS level I: 29, II: 13, III: 15, IV: 3. Distribution of the children with respect to age, spastic subtype (US-CP and BS-CP) and GMFCS level for answers on the CSP-2© and on the two questions regarding maintaining posture and sound sensitivity are presented in Table 1.

### 3.2. Child Sensory Profile-2©

The Child Sensory Profile-2© was answered by 55/60 parents; four parents did not answer due to difficulties understanding the language and one did not hand in the questionnaire. The children with BS-CP were older compared to the US-CP group (*p* = 0.008), and the children at GMFCS levels III-IV were older compared to children at GMFCS level I (*p* = 0.002).

On the sections and quadrants of the CSP-2© (Figure 1), the mean (SD) scores for the entire group of children with CP (*n* = 55) and with respect to the subgroups US-CP (*n* = 18) and BS-CP (*n* = 37), compared to Scandinavian norm-referenced values, are presented in Table 2. The number (percent) of children with scores >2 SD are presented in Table 3. The entire group of children with CP had significantly higher mean scores on all sections and quadrants compared to norm-referenced values (Table 2).

There was a low correlation between age and the scores in the quadrant of Seeking (r = −0.305, *p* = 0.02) and the sensory section of Auditory (r = −0.306, *p* = 0.02) (Table 4).

A score > 2 SD in one or more of the sections or quadrants was found in 30 (55%) of the children (Table 3). In the section of Body Position, 22 (40%) children had scores > 2 SD. In the behavioural section of Conduct, no child had a score > 2 SD (Table 3).

With respect to spastic subtype, higher mean scores on all sections and quadrants compared to the norm were reported in the children with US-CP (*n* = 18) and similarly in the children with BS-CP (*n* = 37) except for the sections of Visual and Touch and the quadrant of Seeking (Table 2). Between the subtype groups, there were no differences in median scores or proportions of scores > 2 SD (proportions are reported in Table 3).

With respect to motor functioning, namely GMFCS levels, there were no differences in median scores or proportions of scores > 2 SD (proportions are reported in Table 3).

### 3.3. Maintaining Body Posture and Sound Sensitivity

The two questions regarding (a) maintaining body posture and (b) sound sensitivity were answered by 57 parents. Two parents answered only question (a) and two parents answered only question (b); consequently, each individual question had 55 answers.

On the question regarding maintaining an erect body posture, 26 (47%) of the children with CP were frequently requested to sit/stand up straight. On the question regarding sound sensitivity at a younger age, 30 (55%) of the children with CP were reported to be sensitive to sounds at a younger age. With respect to spastic subtype, there were no significant differences on any of the questions. With respect to motor functioning, a greater proportion of children at GMFCS levels III-IV compared to level I (14/17 versus 6/26, *p* < 0.001) were frequently requested to sit/stand up straight and were sensitive to sounds at a younger age (14/17 versus 10/26, *p* = 0.005) (Figure 2).

## 4. Discussion

In agreement with our hypothesis, children with spastic CP were reported to have altered responses to sensory experiences, and more than half of the group of children (55%) showed engagement in behaviours in responses to sensory events that might interfere with daily life. Further analysis of the results from the caregiver questionnaire CSP-2© indicated that some children miss sensory input and have responses of concern related to body positions, overrepresented by the BS-CP group. Moreover, results from two additional questions, indicated that attention should be paid to sensory aspects of movement and posture in children at higher GMFCS levels, including children with BS-CP.

Awareness of sensations were reported to differ in one out of four children, who had scores > 2 SD in the sensory processing pattern of Registration in the CSP-2©. These children often miss or lack awareness of sensory stimuli and might react slowly to rapidly presented or low intensity stimuli. Consequently, they may adopt a bystander role during various activities [38]. Moreover, some children in this study were reported to be bothered by sensory input; they had scores > 2 SD in the sensory processing pattern of Avoiding. These children might find it useful to create a structure to avoid unfamiliar sensory experiences [38]. Different sensory processing patterns co-occur in all individuals based on activity demands and contexts. The sensory processing patterns Registration and Avoiding are contrasting in both neurological threshold and self-regulation [38]. However, the finding of this study that the children with CP had high scores in these patterns is in line with previously published research [34,36]. Moreover, in a study in which parents of children with US-CP replied to the CSP-2©, high scores in the quadrants of Registration and Avoiding were associated with difficulties with functional performance as measured by the caregiver assessment Pediatric Evaluation of Disability Inventory [34].

Awareness and registration of sensations is essential for detecting changes in joint and muscle positions [2], and in the CSP-2©, the scores in the section of Body Position contribute to the scores in the quadrant of Registration [38]. In our study, as many as forty percent of the parents reported that their child showed responses “much more than others” (> 2 SD) in the section of Body Position, including statements about propping to support self and becoming tired easily when holding the body in one position. This finding indicates that difficulties with body positions in children with CP may be related to sensory processing, more so in children with BS-CP, who had a high proportion (49%) of scores > 2 SD in this section. Traditionally, the functioning of children with CP has focused on motor disorders, but there is growing recognition of the sensory aspects of motor functioning [29], and sensory disturbances have especially been observed in children with BS-CP at higher GMFCS levels [27].

Differences in executive functioning and attentional and social emotional problems have been reported in children with CP [43]. In our study, several parents reported that their child showed responses “much more than others” in the behavioural sections of Social Emotional and Attentional processing. Previous research on children with CP indicates that high scores in the behavioural section of the Sensory Profile™ might have a negative impact on participation in the school environment, psychological well-being, self-care abilities and social functioning [34,35]. Psychiatric or behavioural problems such as autism, ADHD and mood disorders are described as accompanying disorders in CP [6]. However, an encouraging discovery of this study was that no child had responses “much more than others” in the section of Conduct that measured responses to expectations and included statements on temper tantrums.

The CSP-2© is not specifically developed for children with motor disorders. Parents to eight children (15%) answered “does not apply” (a score of zero) on half or more of the items in the CSP-2©. The children were four boys and four girls with a median age of 15.0 years, all with BS-CP and functioning at GMFCS level: I: 2; II: 3 and III: 3. Despite these reports, the entire group of children with CP had higher mean scores in all sections and quadrants of the CSP-2© compared to norm-referenced values. This underlines the finding that children with CP differ in their responses to sensory events. In the section of Movement, including questions related to balance, surprisingly few children had responses “much more than others” compared to in the section Body Position. One explanation could be that the items are not formulated to reflect the movement disorders of children with CP. One could also speculate that the difficulties with movements in children with CP may be related less to sensory functioning than difficulties with body positions and posture. It has previously been reported that a group of children with BS-CP could perform rather complex movements, but difficulties become obvious as postural demands increased [44].

In the quadrant of Seeking and the sensory sections of Visual and Touch of the CPS-2©, the scores did not differ from the norm-referenced values for the children with BS-CP, a result that was somewhat unexpected since accompanying disorders related to sensations, such as vision and touch, are commonly described in children with CP [6,16,18]. One explanation could be that the items in the CPS-2© are developed to measure behaviours in response to sensory events and not the impairments per se. Another explanation could be the impact of age, which has been reported to influence the responses in these sections [38,39]; the children with BS-CP in this study were older compared to the children in the norm-referenced group and the group of children with US-CP.

In our study, age correlated with the scores in the quadrant of Seeking and the sensory section of Auditory, with older children having lower scores, implying more typical scores. However, the mean scores were still high for the entire group of children with CP compared to norm-referenced values. Consequently, being older did not eliminate the high frequency of responses to sensory events reported.

It has been recommended that results from CSP-2© should be interpreted in conjunction with other sources of information, to get a more comprehensive understanding of the child [38]. Hence, supplementary information was collected through two questions that were formulated based on previous publications and the researchers’ professional experience working with children with CP [20,28]. For the question regarding the ability to maintain an erect body posture, almost half of the parents (47%) replied that their children were frequently requested to sit/stand up straight. Results further indicated that difficulties maintaining posture were more frequent in children at higher GMFCS levels, including children with BS-CP. It is known that neurological disorders affecting the control of movement and posture, such as in CP, alter motor automaticity [45]. Previously, children with CP with difficulties automatically correcting their body posture have been considered as less attentive to sensory input from their body and from the environment. It has been hypothesized and suggested that these children could benefit from physical, verbal or visual cues to guide posture [28]. Difficulties with body positions and maintaining an erect posture could be explained by muscle weakness, which has been related to crouch gait, and the requirement for support when walking [46,47]. However, it has also been suggested that disturbances in spatial orientation and/or perception contribute to the difficulties in maintaining posture in relation to gravity and to motor function observed in some children with BS-CP [20,21,48]. In a previous study from our research group, we found that lower limb muscle strength was not an explanatory factor for variations in standing when 25 children with BS-CP were investigated [21].

On the question regarding sensitivity to sound, more than half (55%) of the parents reported that their children were sensitive to sounds at a younger age, more so at GMFCS levels III-IV. Previously, unexpected sounds have been described to elicit physical signs of perceptual disorders such as the startle reaction [26] and, if present at a young age, these signs have been found stable over time [49]. Some of the parents in this study reported that sudden or loud sounds could elicit behaviours in their child such as jerking/startling, crying and covering the ears with the hands. In-depth descriptions from caregivers of the children’s behaviours and the impact on the functioning and participation in everyday life could have added valuable information.

One of the strengths of this study is that we used parents’ reports reflecting the children’s daily life. Observing and describing behavioural responses in the natural environment is a convenient, non-invasive, accessible method that is valuable for predicting function and promoting participation [50]. Moreover, we utilized norm-referenced scores available in the Swedish version of the CSP-2© [38] for comparison. These scores are based on replies from 729 parents of children in Sweden, Norway and Denmark [39]. As an additional benefit, the Swedish version of the CSP-2© is cross-culturally adapted to maintain content validity at a conceptual level across different cultures [39].

There are limitations of this study, such as the small subgroup samples and the age differences between the subgroups. The maximal age of the children in this study exceeded the maximal age for which the CSP-2© was developed. Despite access to age-appropriate versions of the questionnaire, we chose the same questionnaire for all, due to the relatively small number of participants. One must also be aware that observational assessment of responses to sensory events can describe symptoms, such as the presence of adaptive responses, without describing the underlying causes. Parental assessments may also be influenced by their expectations of the child, which might vary depending on attitudes, traditions, values and previous life experiences in the family [51].

Results from our study showed definite differences in responses (much more than others) to sensory events in daily life in some children with CP. The findings also indicated sensory concerns related to posture in children at higher GMFCS levels, including children with BS-CP. There is a need for the development of screening instruments to identify children with CP that have sensory issues affecting motor functioning. Enhanced understanding of the functioning of the child may enable individualized interventions aiming to promote environmental adaptations, motor development and participation in daily life.

## Figures and Tables

**Figure 1 children-10-01139-f001:**
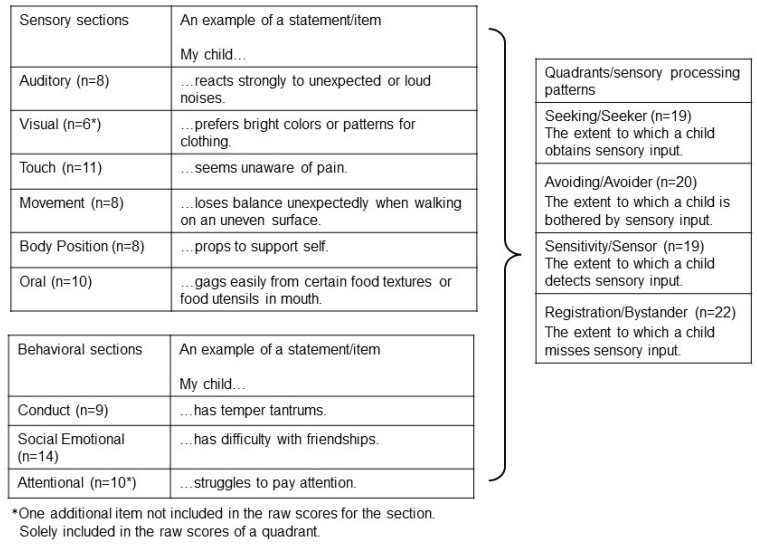
The structure of the Child Sensory Profile-2, CSP-2©, number (*n*) of items for the sections and quadrants and an example of an item in the sections. Short explanation of each sensory processing pattern in the quadrants.

**Figure 2 children-10-01139-f002:**
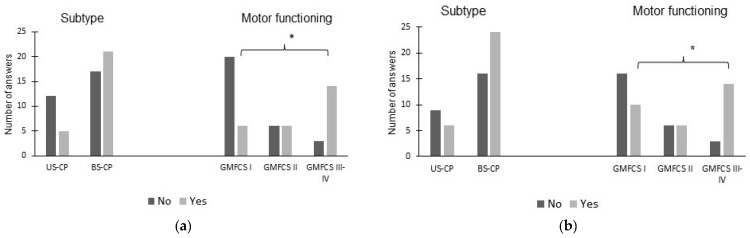
Parents´ answers, yes or no, with number of answers (*n*) on two questions: (**a**) ‘Is your child frequently requested to stand/sit up straight?’ and (**b**) ‘Was your child sensitive to sounds at a younger age?’ Presented with respect to cerebral palsy subtype (unilateral, US-CP, and bilateral, BS-CP) and motor functioning (Gross Motor Function Classification System, GMFCS) levels, with * indicating significant differences (*p*-level ≤ 0.05).

**Table 1 children-10-01139-t001:** Number (*n*) and age (median [min, max] years) of children with spastic cerebral palsy in the entire group and assessed with the Child Sensory Profile-2© (CSP-2©) and the two questions: question (a) ‘Is your child frequently requested to stand/sit up straight?’ and question (b) ‘Was your child sensitive to sounds at a younger age?’ with respect to subtype: unilateral (US-CP) and bilateral (BS-CP), and motor functioning: Gross Motor Function Classification System (GMFCS) levels.

	CSP-2 (*n* = 55)	Question (a) (*n* = 55)	Question (b) (*n* = 55)
US-CP/BS-CP (*n*)	18/37	17/38	15/40
GMFCS levels (*n*)			
I	28	26	26
II	12	12	12
III	13	14	14
IV	2	3	3
Age, median [min, max] years for:			
Spastic CP	12.3 [6.3, 18.0]	12.7 [6.3, 18.0]	12.4 [6.3, 18.0]
US-CP	9.7 [6.3, 17.7]	9.6 [6.3, 17.7]	9.6 [6.3, 17.7]
BS-CP	14.0 [6.4, 18.0]	14.2 [6.4, 18.0]	13.9 [6.4, 18.0]
GMFCS levels:			
I	10.1 [6.3, 18.0]	10.2 [6.3, 18.0]	10.2 [6.3, 18.0]
II	12.1 [6.4, 17.8]	12.1 [6.4, 17.8]	11.0 [6.4, 17.8]
III–IV	15.6 [9.6, 17.7]	15.0 [9.6, 17.7]	15.0 [9.6, 17.7]

**Table 2 children-10-01139-t002:** Child Sensory Profile-2 (CSP-2©) mean (SD) scores for the Scandinavian norm group, the group of children with cerebral palsy (CP) and with respect to subtype: unilateral (US-CP) and bilateral (BS-CP). A two-sided Z-test was used to calculate z-values and *p*-values between the groups of children with CP, US-CP and BS-CP, respectively, compared to the norm group. * Indicates significant differences (*p*-level ≤ 0.05).

CSP-2©	Norm Group Mean (SD)	CP *n* = 55 Mean (SD)	*z*-Value	*p*-Value	US-CP *n* = 18 Mean (SD)	*z*-Value	*p*-Value	BS-CP *n* = 37 Mean (SD)	*z*-Value	*p*-Value
Quadrants/sensory processing patterns:										
Seeking	22.4 (16.7)	29.0 (15.1)	2.9	0.003 *	34.4 (13.1)	3.1	0.002 *	26.4 (15.5)	1.5	0.144
Avoiding	26.3 (17.7)	42.9 (18.0)	7.0	<0.001 *	42.7 (17.7)	3.9	<0.001 *	43.0 (18.3)	5.7	<0.001 *
Sensitivity	21.6 (15.0)	32.3 (13.8)	5.3	<0.001 *	34.8 (10.5)	3.7	<0.001 *	31.1 (15.1)	3.9	<0.001 *
Registration	22.4 (17.3)	46.3 (16.3)	10.2	<0.001 *	47.0 (14.6)	6.0	<0.001 *	46.0 (17.2)	8.3	<0.001 *
Sensory sections:										
Auditory	14.3 (7.5)	18.8 (8.8)	4.4	<0.001 *	20.5 (8.4)	3.5	<0.001 *	17.9 (9.0)	2.9	0.003 *
Visual	9.2 (5.2)	11.5 (6.0)	3.3	<0.001 *	12.7 (4.2)	2.9	0.004 *	10.9 (6.6)	1.9	0.052
Touch	10.8 (8.7)	14.3 (8.2)	3.0	0.003 *	16.6 (6.9)	2.8	0.005 *	13.2 (8.6)	1.7	0.091
Movement	7.9 (6.8)	13.1 (6.4)	5.7	<0.001 *	14.9 (6.4)	4.4	<0.001 *	12.2 (6.3)	3.9	<0.001 *
Body position	6.5 (6.6)	18.4 (7.6)	13.4	<0.001 *	16.9 (6.0)	6.7	<0.001 *	19.1 (8.3)	11.6	<0.001 *
Oral	9.8 (9.3)	14.1 (9.2)	3.4	<0.001 *	15.1 (7.0)	2.4	0.016 *	13.7 (10.1)	2.5	0.012 *
Behavioural sections:										
Conduct	10.9 (8.4)	15.2 (5.9)	3.8	<0.001 *	16.2 (5.0)	2.7	0.008 *	14.8 (6.3)	2.8	0.005 *
Social emotional	16.9 (13.9)	31.7 (13.0)	7.9	<0.001 *	30.0 (13.0)	4.0	<0.001 *	32.6 (13.1)	6.9	<0.001 *
Attentional	10.9 (8.9)	19.5 (9.8)	7.2	<0.001 *	22.0 (8.8)	5.3	<0.001 *	18.2 (10.1)	5.0	<0.001 *

**Table 3 children-10-01139-t003:** Distribution, number (*n*) and percent (%) of children scoring > 2 SD of the norm-referenced values on the quadrants and sections and with a score > 2 SD in one or more sections and/or quadrants of the Child Sensory Profile-2©) in the group of children with cerebral palsy (CP) and with respect to unilateral (US-CP) and bilateral spastic CP (BS-CP) and level of Gross Motor Function Classification System (GMFCS). A Pearson Chi-Square Test was used to calculate differences in proportions of children in the subgroups. Significance level, *p* ≤ 0.05.

Child Sensory Profile-2©	CP *n* = 55	US-CP *n* = 18	BS-CP *n* = 37	Differences between US-CP/BS-CP	GMFCS I *n* = 28	GMFCS II *n* = 12	GMFCS III–IV *n* = 15	Differences between GMFCS
Quadrants:	*n* (%) Score >2 SD	*n* (%) Score >2 SD	*n* (%) Score >2 SD	*p*-Value	*n* (%) Score >2 SD	*n* (%) Score >2 SD	*n* (%) Score >2 SD	*p*-Value
Seeking	1 (2)	1 (6)	0 (0)	0.148	1 (4)	0 (0)	0 (0)	0.612
Avoiding	8 (15)	4 (22)	4 (11)	0.260	4 (14)	1 (8)	3 (20)	0.693
Sensitivity	5 (9)	1 (6)	4 (11)	0.525	2 (7)	1 (8)	2 (13)	0.793
Registration	14 (26)	4 (22)	10 (27)	0.701	5 (18)	3 (25)	6 (40)	0.283
Sensory sections:								
Auditory	7 (13)	4 (22)	3 (8)	0.141	5 (18)	2 (17)	0 (0)	0.221
Visual	3 (6)	0 (0)	3 (8)	0.214	1 (4)	0 (0)	2 (13)	0.260
Touch	3 (6)	2 (11)	1 (3)	0.198	2 (7)	0 (0)	1 (7)	0.641
Movement	5 (9)	3 (17)	2 (5)	0.173	2 (7)	1 (8)	2 (13)	0.793
Body position	22 (40)	4 (22)	18 (49)	0.061	8 (29)	5 (42)	9 (60)	0.133
Oral	5 (9)	1 (6)	4 (11)	0.525	2 (7)	1 (8)	2 (13)	0.793
Behavioural sections:								
Conduct	0 (0)	0 (0)	0 (0)	ns	0 (0)	0 (0)	0 (0)	ns
Social emotional	10 (18)	4 (22)	6 (16)	0.588	5 (18)	1 (8)	4 (27)	0.470
Attentional	8 (15)	3 (17)	5 (14)	0.756	4 (14)	3 (25)	1 (7)	0.405
Children with a score > 2 SD in one or more sections and/or quadrants	30 (55)	7 (39)	23 (62)	0.104	13 (46)	6 (50)	11 (73)	0.225

**Table 4 children-10-01139-t004:** Correlations between the scores on each of the quadrants and sections in the Child Sensory Profile-2© and age calculated with Spearman’s correlation coefficient (r_s_). * Indicates significant correlations (*p* ≤ 0.05 and r_s_ ≤ −0.30 or ≥ 0.30).

	Correlation Coefficient (r_s_)	*p*-Value
Child Sensory Profile-2©	Age	
Quadrants:		
Seeking	−0.305 *	0.024 *
Avoiding	-0.006	0.968
Sensitivity	-0.109	0.429
Registration	0.041	0.765
Sensory sections:		
Auditory	−0.306 *	0.023 *
Visual	−0.014	0.918
Touch	−0.079	0.569
Movement	−0.133	0.333
Body Position	0.076	0.579
Oral	−0.235	0.085
Behavioural sections:		
Conduct	−0.223	0.101
Social Emotional	0.106	0.442
Attentional	−0.077	0.577

## Data Availability

The datasets generated and analysed during the current study are available from the corresponding author on reasonable request.

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
