# Peer review of "Responses to Sensory Events in Daily Life in Children with Cerebral Palsy from a Parent Reported Perspective and in a Swedish Context"

_children, 2023, doi:10.3390/children10071139_

Round 1

Reviewer 1 Report

Excellent manuscript.

One suggestion, please briefly describe demographics of the patients (GMFCS level, age, gender) for the 15 % of parents that responded 'does not apply' to half or more of the questions mentioned on page 8 lines 291-293. As a clinician, I believe this would be beneficial information.

Author Response

Thank you for the overall positive response to the manuscript. See enclosed document for our response to your suggestion.

Reviewer 2 Report

1. Did the fact that more patients with better level pf GMFCs influence outcome?

2. Parents need description of socioeconomic background, number of children, help within the household, level of occupation and education level

3. Conclusion worse GMFCS level 3/4 request sit stand compared better GMFCS level does not make sense

4. Are bilateral CP quads or mixed ataxic/ spastic/

English ok

Author Response

Thank you for reviewing the manuscript. Please see our response to your comments in enclosed document. 

Reviewer 3 Report

This paper is an interesting study of the sensory profile of children with cerebral palsy.

 Since some data are described only in the text and not in the tables, they are difficult for the reader to recognize.

In particular, the correlation between age and the results of CSP-2 is an important matter when interpreting the results, so it is necessary to describe the results in table.

 The data on the number of children with scores >2SD in one or more of the sections or quadrants were not presented in Table 3. I suggest adding these data and p value in table 3.

Author Response

Thank you for your positive comments and valuable suggestions. Please see attachment for our response. 
